# Effects of ankle Kinesio taping on knee and ankle joint biomechanics during unanticipated jumps in collegiate athletes

Quan Liu[1,2☺], Ling Wang[2,3☺], Fan Dai[1], Guanglan Wang[3‡]*, Peng Chen[4‡]*

1 Wuhan Fourth Hospital, Wuhan, Hubei Province, China, 2 Key Laboratory of Sports Engineering of General Administration of Sport of China, Wuhan Sports University, Wuhan, Hubei Province, China, 3 School of Sports Medicine, Wuhan Sports University, Wuhan, Hubei Province, China, 4 School of Exercise and Health, Shanghai University of Sport, Shanghai, China

☺ These authors contributed equally to this work.
‡ These authors contributed equally to this work and share corresponding authorship
* pengchen7173@163.com (PC); 1608175918@qq.com (GW)

## Abstract

### Objective

Most biomechanical research on the application of Kinesio taping (KT) to the ankle joint focused on testing anticipated movements. However, ankle sprains frequently occur in real life in unanticipated situations, where individuals are unprepared and face sudden external stimuli. This situation is completely different from the anticipated situation. The aim of the present study was to investigate the effects of ankle KT application on the kinematic and kinetic characteristics of the knee and ankle joints during unanticipated jump tasks in collegiate athletes.

### Methods

Eighteen healthy collegiate athletes experienced three taping conditions in a randomized order: no taping (NT), placebo taping (PT), and KT, and performed unanticipated jump tasks. A 9-camera infrared high-speed motion capture system was employed to collect knee and ankle kinematic data, and a 3-dimensional force plate was utilized to collect knee and ankle kinetic data during the tasks.

### Results

During the right jumps, KT significantly increased peak knee flexion angle ($P = 0.031$) compared to NT and significantly decreased peak vertical ground reaction force ($P < 0.001$, $P = 0.001$) compared to NT and PT. During the left jumps, KT significantly reduced peak ankle inversion angle ($P = 0.022$, $P < 0.001$) and peak ankle inversion moment ($P = 0.002$, $P = 0.001$) compared to NT and PT.

**Data Availability Statement:** All relevant data are within the manuscript and its Supporting Information files.

**Funding:** The author(s) received no specific funding for this work.

**Competing interests:** The authors have declared that no competing interests exist.

## Conclusion

During unanticipated jump maneuvers, KT reduced peak ankle inversion angle, peak vertical ground reaction force, and peak ankle inversion moment and increased peak knee flexion angle in collegiate athletes.

## 1. Introduction

Globally, approximately 1 in 10,000 people suffer ankle sprains daily, with 74% of these cases eventually progressing to chronic ankle instability, which poses a substantial healthcare burden [1,2]. Therefore, ankle sprain prevention is not only crucial for individual well-being but also holds significant implications for both the healthcare and economic aspects of society. Kinesio taping (KT), a highly elastic and extensible sports taping, is widely used in ankle sprain prevention due to its ability to stabilize joints, reduce impact forces, and modulate muscle activity [3,4]. Extensive research has been conducted on the effect of KT on ankle sprains in healthy people. For instance, Yu et al. [5] reported that KT significantly improved ankle inversion proprioception in healthy participants, which facilitated better control of the ankle joint and reduced inversion sprains during landing. Chang et al. [6] found that KT could improve balance function in healthy collegiate athletes and did not restrict ankle dorsiflexion range of motion.

However, most studies primarily focused on the biomechanical characteristics of KT on anticipated movements, although it is worth noting that unanticipated movements are more prevalent in actual athletic competitions [7,8]. Research indicated that the risk of lower extremity injury is significantly higher in unanticipated circumstances than in anticipated conditions [9,10]. This may be attributed to the fact that anticipated movements lead participants to predict movements in advance and thereby adapt neuromuscular control strategies to prevent injury, ultimately confounding real biomechanical changes [9,11]. During unanticipated conditions, the body elicits a startle reflex, which results in a series of widespread, transient, and involuntary neuromuscular activity changes [12]. This alteration may increase the risk of ankle sprains by preventing the body from promptly adjusting its movements and absorbing the impact of landing [13].

Single-leg jump landing is considered a high-risk maneuver for ankle sprains due to the large impulsive forces involved [14]. During a jump landing task, the lower extremities are tasked not only with supporting the weight of the body but also with mitigating the impact forces of landing. A study showed that 45% of ankle sprains in basketball players occur during landing [15]. Given that the lower limbs operate as an integrated entity, changes in a joint's movement pattern invariably trigger modifications in the neighboring joints [16]. Theisen et al. [17] reported that patients with chronic ankle instability have reduced knee flexion angles compared with the healthy control group, implying a higher risk of anterior cruciate ligament (ACL) injury. The study by Kramer et al. [18] confirmed a significant correlation between a history of ankle sprains and the risk of ACL injuries. Therefore, it is also important to explore the effect of ankle KT on knee joint biomechanics for the prevention of ACL injuries. However, the current studies have primarily focused on investigating the effects of KT on ankle joint biomechanical characteristics.

The present study employed a randomized signal light system emitting three distinct signals to simulate an unanticipated environment similar to actual movement. We compared the differences in knee and ankle biomechanical characteristics among collegiate athletes under three

treatments: no taping (NT), placebo taping (PT), and KT, thus providing a reference for the application of KT in actual sports. The hypothesis of this study was that KT can reduce peak ankle plantarflexion and inversion angles and moments, and increase peak knee flexion angle.

## 2 Materials and methods

### 2.1 Participants

The appropriate sample size for this study was determined using a priori power analysis (G*power 3.1.2), with an effect size of 0.35, a statistical power of 0.8, a significance level of 0.05, and a dropout rate of 0.15. The analysis revealed a requisite minimum of 18 participants. Eighteen collegiate athletes from Wuhan Sports University (11 males and 7 females; age, 23.56 ± 2.41 years; height, 171.23 ± 8.74 cm; weight, 63.44 ± 12.11 kg) were recruited between October 14, 2023 and October 28, 2023. The right leg was the dominant leg in all participants. The inclusion criteria were as follows: (1) engaging in a certain sport (basketball, football, or volleyball), (2) Tegner score ≥7 [19], and (3) exercise frequency ≥3 times per week, and exercise duration ≥30 min per session [20]. The exclusion criteria were as follows: (1) presence of lower limb fracture or surgical history, (2) history of lower limb sports injury within the past 6 months, and (3) allergic reactions to KT. The study was approved by the local ethics committee (approval no. KY2023-069-03), and all participants were informed of the experimental procedure and provided their written informed consent.

### 2.2 Taping procedure

KT was administered utilizing 5 cm × 5 m Kindmax taping, and the taping methodology employed was consistent with Lin et al.'s study, with a tension level set at 50% [21]. This tension was determined by the taping length calculated by modifying the formula proposed by Nunes et al. [22] The formula for taping length was as follows:

$$\text{Cutting length} = \left( \frac{\text{actual length} - 8 \text{ cm}}{1.5} + 8 \text{ cm} \right) \times 1.1$$

Herein, 8 cm represents the combined anchor and tail length, and 1.1 denotes the pretension of the taping. The taping methods used were as follows: (1) Tibialis anterior: Employing the I-type taping method, the participants assumed a supine position with the ankle in plantarflexion. The anchor was secured to the tibial tuberosity, and the tail was secured to the anterior part of the foot. The taping covered the tibialis anterior muscle belly. (2) Peroneus longus: Employing the I-type taping method, the participants assumed a supine position with the ankle in inversion. The anchor was attached to the fibular head, and the tail was attached to the medial malleolus. The taping covered the peroneus longus muscle belly, bypassing the sole of the foot to the medial malleolus. (3) Gastrocnemius: Employing the Y-type taping method, the participants assumed a prone position with the ankle in dorsiflexion. The anchor was fixed to the sole of the foot, and the tails were fixed to the medial and lateral epicondyles of the femur. The taping covered the gastrocnemius muscle belly (**Fig 1**). PT followed the same taping procedure as KT but without applying tension, whereas NT received no intervention. The taping of the dominant leg for all participants was administered by a professional taping specialist, with the dominant leg defined as the preferred leg for kicking [23]. All participants received three different interventions in randomized sequence with a 1-w interval between each condition.

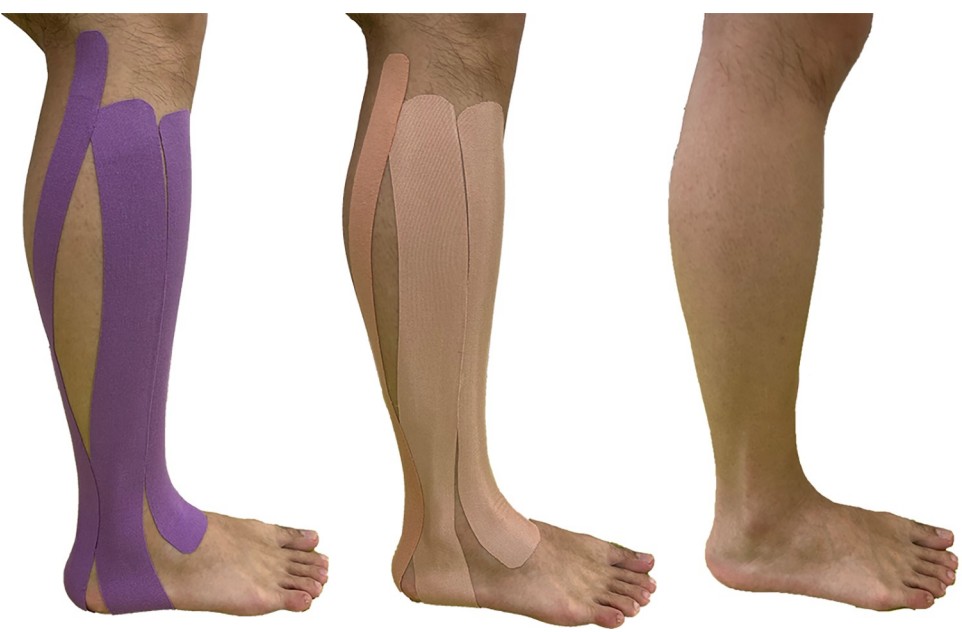

**Fig 1. Interventions under three conditions.** (A) Kinesio taping, (B) placebo taping, (C) no taping.

## 2.3 Data collection

Participants wore tight-fitting clothing and were barefoot. After a 10-min warm-up, they received the interventions. Thirty-eight reflective markers were affixed according to the modified Helen Hayes model [24]. The participants then performed unanticipated jumps, and the jump tasks were modified from the study of Mclean et al [25]. The starting point was 2 m behind the force plate, and the participants executed the task based on the color of a signal light that was 2.5 m in front of the force plate. The signal light activated immediately (within 1 s) following the beginning of the participants' running. When the signal light turned blue, the participants ran and landed on the right force plate with the right leg and quickly jumped to the left force plate. When it turned green, they ran and landed on the left force plate with the right leg and then jumped to the right force plate. When the light was red, the participants ran, landed each leg on two separate plates, and performed a vertical jump (**Fig 2**). Along the direction of the runway, a pair of infrared grating timers was positioned 50 cm ahead of the front edge of the first force plate and 10 cm behind it. The participant's velocity was calculated based on the distance between the two sets of gratings and the time taken to complete it. Prior to formal testing, the participants completed 6–10 running practices to ensure they could run at the speed required for the present study (2.7 ± 0.4 m/s) during subsequent testing. The participants were tasked with 9 jumps for each intervention condition, comprising 3 left jumps, 3 right jumps, and 3 double-leg jumps. Both the intervention and jump sequences were generated using the Excel random function to ensure that the participants were blinded to them. Kinematics and kinetics data were synchronously recorded utilizing a 9-camera infrared high-speed motion capture system (sampled at 200 Hz; Vicon T40, Oxford, UK) and a 3-dimensional force plate (sampled at 1000 Hz; Kistler, Winterthur, Switzerland). Single-leg jump landing is considered a high-risk maneuver for ankle sprains due to the large impulsive forces involved [14]. Therefore, only the single-leg jumping maneuver was selected for between-group analysis in the present study. The double-leg jumping was used to increase the jumping

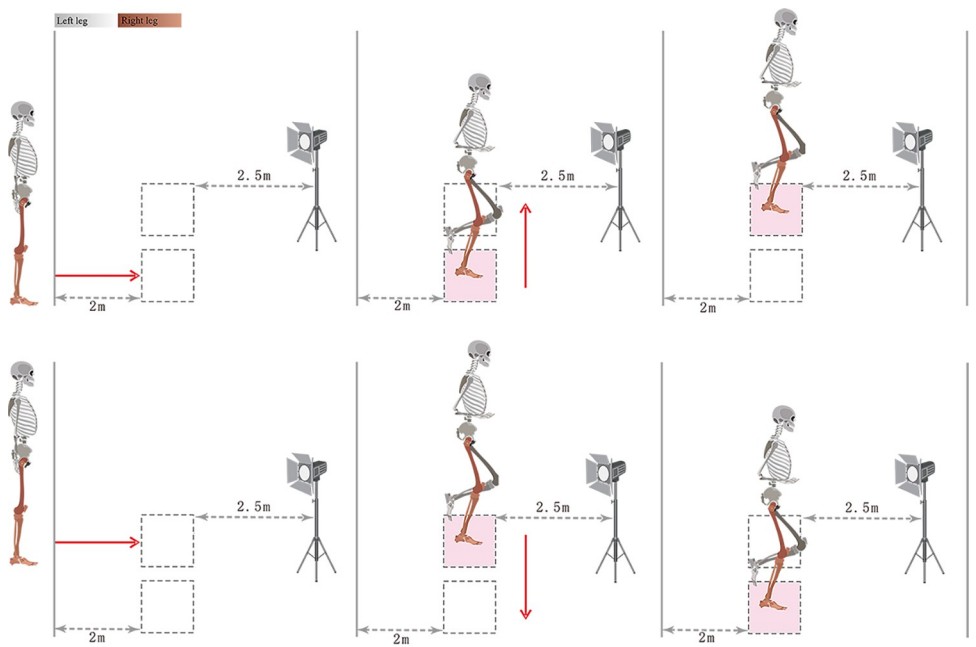

**Fig 2. Unanticipated single-leg jump technique.** (A) Left jump, (B) Right jump; Solid arrows denote the direction of movement.

pattern and avoid the influence of anticipation resulting from predicting jumping actions, thereby changing the biomechanical characteristics.

## 2.4 Data processing

Kinematic and kinetic data of the right knee and ankle during the early landing phase (0–200 ms after initial contact) of the jump movement were collected and normalized into 41 data points [26]. Kinematic data included peak knee flexion, valgus, internal rotation angles and corresponding peak angular velocities; peak ankle plantarflexion, inversion angles and corresponding peak angular velocities; and time to peak vertical ground reaction force (vGRF). Kinetic data consisted of peak vGRF; peak knee flexion, valgus, and internal rotation moments, as well as peak ankle plantarflexion and inversion moments. Data processing and calculations were performed in Visual 3D software (C-Motion, Germantown, Maryland, USA). A low-pass Butterworth digital filter with a cutoff frequency of 12 Hz was employed to filter the kinematic and kinetic data. Joint angles and angular velocities were calculated for each distal segment relative to the proximal segment. Kinetic data were obtained by inverse dynamic methods and normalized to the participant's body weight. The moment of initial contact was defined as vGRF exceeding 50 N [27].

## 2.5 Statistical analysis

Statistical analyses were performed using SPSS 26.0 software (SPSS Inc., Chicago, IL, USA). The data were normally distributed according to the Shapiro–Wilk test and were expressed as mean ± standard deviation. One-way repeated measures ANOVA was conducted to explore the effect of the different interventions (NT, PT, and KT) on the kinematic and kinetic data of the knee and ankle joints. Bonferroni post-hoc test was employed for pairwise comparisons. $P < 0.05$ was considered statistically significant.

**Table 1. Kinematic characteristics.**

| Outcome | Left jumps | | | Right jumps | | |
|---|---|---|---|---|---|---|
| | NT | PT | KT | NT | PT | KT |
| Peak knee flexion angle (°) | 39.82±6.32 | 39.98±6.51 | 40.10±5.15 | 39.38±6.78 | 39.66±5.31 | **41.33±5.63**<sup>*</sup> |
| Peak knee valgus angle (°) | 5.98±3.97 | 5.60±3.71 | 5.36±4.03 | 5.65±2.43 | 5.42±2.44 | 5.23±2.49 |
| Peak knee internal rotation angle (°) | 7.34±4.01 | 6.95±4.25 | 5.93±3.55 | 8.67±3.72 | 7.75±3.81 | 7.51±3.05 |
| Peak ankle plantarflexion angle (°) | 25.56±8.29 | 23.56±9.94 | 24.33±7.44 | 28.51±6.38 | 27.19±7.91 | 28.42±6.18 |
| Peak ankle inversion angle (°) | 4.38±2.92 | 4.55±2.20 | **3.07±2.05**<sup>*#</sup> | 5.79±2.56 | 6.15±2.74 | 6.50±3.81 |
| Peak knee flexion angular velocity (°/s) | 396.18±61.51 | 420.07±97.03 | 395.46±81.12 | 366.88±104.97 | 349.81±104.36 | 352.44±100.25 |
| Peak knee valgus angular velocity (°/s) | 75.72±21.58 | 75.21±22.28 | 75.61±23.04 | 82.85±32.88 | 72.47±38.51 | 73.35±35.53 |
| Peak knee internal rotation angular velocity (°/s) | 141.64±61.56 | 122.02±33.13 | 122.97±30.95 | 99.04±28.02 | 91.60±39.11 | 103.46±26.80 |
| Peak ankle plantarflexion angular velocity (°/s) | 92.42±41.39 | 89.86±33.47 | 91.35±33.44 | 86.42±27.90 | 85.60±24.52 | 85.50±24.22 |
| Peak ankle inversion angular velocity (°/s) | 57.68±32.68 | 66.23±31.27 | 51.87±24.77 | 73.53±19.36 | 72.67±19.58 | 75.99±26.61 |
| Time to peak vGRF(ms) | 79.72±16.85 | 78.61±14.53 | 86.94±16.19 | 78.06±19.41 | 81.67±20.36 | 87.50±15.55 |

Notes

$^{*}P < 0.05$, compared with NT

$^{\#} P < 0.05$, compared with PT. NT, no taping; PT, placebo taping; KT, Kinesio taping. vGRF, vertical ground reaction force.

## 3 Results

### 3.1 Kinematic characteristics

During the right-jump tasks, KT significantly increased peak knee flexion angle ($P = 0.031$) compared with NT. During the left-jump tasks, KT significantly reduced peak ankle inversion angle ($P = 0.022$, $P < 0.001$) compared to NT and PT. No significant differences in other kinematic outcomes were observed ($P > 0.05$) (**Table 1**).

### 3.2 Kinetic characteristics

During the right-jump tasks, KT significantly decreased peak vGRF ($P < 0.001$, $P = 0.001$) compared to NT and PT. During the left-jump tasks, KT significantly reduced peak ankle inversion moment ($P = 0.002$, $P = 0.001$) compared to NT and PT. No significant differences in other kinetic outcomes were observed ($P > 0.05$) (**Table 2**).

**Table 2. Kinetic characteristics.**

| Outcome | Left jumps | | | Right jumps | | |
|---|---|---|---|---|---|---|
| | NT | PT | KT | NT | PT | KT |
| Peak vGRF(BW) | 1.90±0.16 | 1.91±0.22 | 1.90±0.18 | 1.88±0.16 | 1.85±0.15 | **1.75±0.13**<sup>*#</sup> |
| Peak knee flexion moment (Nm/kg) | 0.45±0.20 | 0.46±0.26 | 0.41±0.22 | 0.47±0.21 | 0.45±0.18 | 0.46±0.16 |
| Peak knee valgus moment (Nm/kg) | 0.17±0.10 | 0.19±0.10 | 0.19±0.11 | 0.94±0.38 | 0.86±0.35 | 0.79±0.39 |
| Peak knee internal rotation moment (Nm/kg) | 0.08±0.03 | 0.08±0.04 | 0.08±0.05 | 0.23±0.14 | 0.24±0.14 | 0.21±0.13 |
| Peak ankle plantarflexion moment (Nm/kg) | 1.42±0.28 | 1.45±0.30 | 1.40±0.21 | 1.17±0.24 | 1.14±0.23 | 1.10±0.23 |
| Peak ankle inversion moment (Nm/kg) | 0.81±0.20 | 0.77±0.15 | **0.66±0.18**<sup>*#</sup> | 0.18±0.13 | 0.17±0.12 | 0.18±0.12 |

Notes

$^{*}P < 0.05$, compared with NT

$^{\#} P < 0.05$, compared with PT. NT, no taping; PT, placebo taping; KT, Kinesio taping. vGRF, vertical ground reaction force; BW, body weight; N, Newton.

## 4 Discussion

The present study aimed to compare the differences in knee and ankle joint biomechanical characteristics in collegiate athletes among the three interventions by simulating unanticipated jumps similar to actual sports. Partially consistent with the expected results, the present study revealed that ankle KT reduced peak ankle inversion angle, peak vGRF, and peak ankle inversion moment and increased peak knee flexion angle.

### 4.1 Kinematics

Excessive plantarflexion places the ankle joint in an unstable, sprain-prone position and increases tension on the lateral ankle ligaments, thereby exacerbating the risk of sprains [28]. Wright et al. [29] established an ankle injury model and concluded that a significant ankle plantarflexion angle upon initial contact extends the ground reaction force arm of the subtalar joint, escalating the susceptibility to ankle sprains. The results of the present study indicated that KT did not significantly reduce peak ankle plantarflexion angle compared to NT and PT. On the contrary, Kuni et al. [30] found that KT significantly decreased peak ankle plantarflexion angle during the landing phase of drop landing experiments in healthy participants compared to NT. We believe that the disparity in the above findings could be related to the participants included in the studies. The participants included in our study were healthy athletes, whereas Kuni et al. [30] focused on the general healthy population. Athletes are potentially superior in physiological and functional capacity to the general population. Therefore, athletes might experience a diminished scope for improvement due to the influence of the "ceiling effect" [31].

Previous research indicated that excessive ankle inversion leads to an increase in talus medial displacement, thereby augmenting tension on the lateral ankle ligaments [32]. This is the primary mechanism leading to ankle sprains [33]. Epidemiological data demonstrated that 70% of ankle sprains are attributed to excessive ankle inversion [34]. The results of the present study showed that KT significantly reduced peak ankle inversion angle in the early landing phase compared to NT and PT. Similarly, Botsis et al. [35] tested the biomechanical characteristics of healthy ballet dancers during an anticipated jump landing and found that KT reduced ankle inversion angle. The decrease in peak ankle inversion angle following KT application in the present study may be because the applied taping covered the peroneus longus muscle belly up to the medial malleolus, effectively providing support to the lateral ankle ligaments, thereby aiding in restricting ankle inversion. Additionally, it may be attributed to the improvement of proprioception by KT, enabling the participants to accurately perceive joint angles, thereby exhibiting reduced ankle inversion angle [5]. Wei et al. [36] also indicated that patients with poor proprioception may be more sensitive to KT, more susceptible to its facilitative effects, and transmit more proprioceptive information from joint structures to the nervous system. As a result, patients with poor proprioception tended to receive more benefits than healthy participants with good proprioception. Although the participants in the present study were healthy athletes, the test movements selected in our study were unanticipated jumps. Considering the lower stability and poorer proprioceptive input of athletes during unanticipated jumps, KT may be more effective in promoting proprioception.

Decreased knee flexion angle is one of the biomechanical risk factors for ACL injuries [37]. Our findings showed that KT application resulted in a significant increase in peak knee flexion angle compared to NT and PT. Given that the taping did not cover the knee joint, the likelihood of KT improving peak knee flexion angle through mechanical effects is relatively low. The result may be attributed to the alteration of ankle joint biomechanical characteristics by KT, thereby indirectly influencing knee joint biomechanical features. Moreover, the taping

may exert psychological effects on the participants. The three tapings used in our study mainly covered the lower leg and the ankle joint, and the presence of the taping and the applied tension might have subjectively made the participants feel the foot and ankle restriction effect. As a result, a compensatory strategy of increased knee flexion angle during landing could have been adopted.

## 4.2 Kinetics

Excessive vGRF not only exacerbates ankle joint loading and increases the risk of ankle sprains but also transmits impact forces to the knee joint, resulting in a significantly elevated risk of soft tissue injuries in the knee joint, particularly ACL [29,38]. The findings of the present study demonstrated that KT significantly reduced peak vGRF. However, Yalfani et al. [39] conducted ankle KT on patients with chronic ankle instability and found that vGRF during lateral landing tasks was significantly higher than that in the control group. The disparities in results could be related to variations in taping techniques. Yalfani et al. [39] adopted a KT application method similar to the basket weaves technique of athletic taping. This approach will mechanically restrict the foot and ankle, thereby reducing the range of motion of the lower limb joints [40]. During the landing phase of the jump movements, vGRF is predominantly buffered through the sagittal plane motion of the lower extremities [41]. Reduced joint range of motion decreases the absorption of vGRF, leading to the observed increase in peak vGRF. The taping technique employed in the present study did not involve full coverage of the foot and ankle, potentially avoiding excessive restriction on lower limb joints. Moreover, the present study indicated an increase in peak knee flexion angle, which can be advantageous for reducing peak vGRF. The strategy of increasing knee flexion angle to buffer vGRF can reduce the demand for the ankle joint to cushion the impact of landing. This can not only decrease the load on lateral ankle ligaments and mitigate the risk of ankle sprains but also may have potential implications for reducing the risk of ACL injuries [42,43].

As the first joint to make contact with the ground, the ankle joint bears substantial loads during jump landing. The present study found that KT reduced peak ankle inversion moment during the landing phase of unanticipated jump tasks. Research indicated that taping the peroneus longus muscle can facilitate muscular activity [44]. In addition, the taping retraction from the medial malleolus to the lateral malleolus imparts tension to ankle eversion, which facilitates the restriction of inversion [3]. This may lead to reduced ankle inversion moment. The findings of the present study also demonstrated that KT can reduce ankle inversion angle, thus providing support for this hypothesis. Furthermore, the reduction in peak vGRF diminished the stress and load on the ankle joint, potentially contributing to a decrease in the ankle inversion moment.

## 4.3 Clinical implications

Most studies primarily focused on the biomechanical characteristics of KT on anticipated movements, although unanticipated movements are more prevalent in actual athletic competitions [7,8]. A study indicated that patients with chronic ankle instability exhibit different biomechanical patterns and muscle activation under unexpected conditions compared to those under expected conditions [45]. Simpson et al. [9] observed that the risk of lower extremity injury is significantly higher in unanticipated circumstances than in anticipated conditions. Therefore, testing unanticipated movements can better reflect the actual biomechanical characteristics than anticipated movements. The results of the present study showed that KT can help to decrease peak ankle inversion angle and moment. This served to ameliorate the risky posture for ankle sprains during landing, reducing the tension and load on the lateral ankle

ligaments. In addition, KT can increase peak knee flexion angle and decrease peak vGRF, which helps to better cushion loads during landing and has a positive effect on reducing the risk of lower extremity injuries. In actual sports training and competitions, KT can be considered an effective tool for mitigating the risk of ankle sprains.

### 4.4 Limitations

The present study has some limitations. First, only healthy individuals without lower limb injuries included in our study. Therefore, the results of our study may not be generalized to individuals with pathology. However, studies in healthy participants may have implications for the prevention of sports injuries. Another limitation is that men and women were not analyzed separately because of the relatively small sample size, despite some evidence of biomechanical differences between sexes [46,47].

### 5 Conclusion

During unanticipated jump maneuvers, KT reduced peak ankle inversion angle, peak vGRF, and peak ankle inversion moment and increased peak knee flexion angle in collegiate athletes. This suggests that KT may have a positive effect on reducing the risk of ankle sprains in healthy individuals. Future research should explore whether KT can also mitigate the risk of ankle sprains in individuals with musculoskeletal disorders under unanticipated conditions.

## Supporting information

**S1 Data. Raw data.**
(ZIP)

## Acknowledgments

The authors would like to thank the Key Laboratory of Sports Engineering of the General Administration of Sport of China, Wuhan Sports University for equipment support and all the participants for their participation in this experiment.

## Author Contributions

**Conceptualization:** Quan Liu, Ling Wang, Peng Chen.

**Data curation:** Quan Liu, Ling Wang, Fan Dai.

**Methodology:** Fan Dai, Guanglan Wang.

**Supervision:** Guanglan Wang.

**Writing – original draft:** Quan Liu, Ling Wang.

**Writing – review & editing:** Quan Liu, Ling Wang, Peng Chen.

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
