## [Decision Letter · Decision Letter 0]

19 Mar 2024

PONE-D-24-02413Effects of Ankle Kinesio Taping on Knee and Ankle Joint Biomechanics During Unanticipated Jumps in Collegiate AthletesPLOS ONE

Dear Dr. Chen,

Thank you for submitting your manuscript to PLOS ONE. After careful consideration, we feel that it has merit but does not fully meet PLOS ONE’s publication criteria as it currently stands. Therefore, we invite you to submit a revised version of the manuscript that addresses the points raised during the review process.

We look forward to receiving your revised manuscript.

Kind regards,

Fei Yan

Academic Editor

PLOS ONE

Journal Requirements:

Reviewers' comments:

Reviewer's Responses to Questions

**Comments to the Author**

1. Is the manuscript technically sound, and do the data support the conclusions?

Reviewer #1: Partly

Reviewer #2: Yes

2. Has the statistical analysis been performed appropriately and rigorously? 

Reviewer #1: Yes

Reviewer #2: Yes

3. Have the authors made all data underlying the findings in their manuscript fully available?

Reviewer #1: Yes

Reviewer #2: Yes

4. Is the manuscript presented in an intelligible fashion and written in standard English?

Reviewer #1: No

Reviewer #2: Yes

5. Review Comments to the Author

Reviewer #1: The manuscript entitled "Effects of Ankle Kinesio Taping on Knee and Ankle Joint Biomechanics During Unanticipated Jumps in Collegiate Athletes", aimed to investigate the effects of ankle KT application on the kinematic and kinetic characteristics of the knee and ankle joints during unanticipated jump tasks. The authors have conducted motion capture analyses to 18 participants and concluded that KT reduced peak ankle inversion angle, peak vertical ground reaction force, and increased peak knee flexion. While the paper addresses an interesting issue, it is not publishable in its current form. There are several critical concerns about the experiment design that greatly undermine the significance of the research outcomes. In addition, the language quality of the manuscript falls short of publication standards. Many sentences in the content are confusing and several arguments are self-contradictory.

General comments

1. Based on the authors’ introduction, “unanticipated movement tasks” appeared to be an important element of the study and served as a key factor that would preassembly influence the research outcome. However, it is very the participants were considered situated in an “unanticipated” scenario using the current research setup as described in the manuscript. I believe that participants were well aware of the jumping tasks before commencement and they were only required to respond to three alternative conditions. Signals were given to the participants before they kicked off while they had the extra time to prepare for the correct tasks, particularly landing, during running on a 2-meter runway. I wonder how much the “unanticipated” element was included in the research outcome.

2. I think the effects of KT on ankle kinematics are well established, that is why it is widely applied in sports and clinics. Without involving other factors in the research setup, the study outcomes added very limited knowledge to the current understanding of the mechanism.

Specific comments

Line 25. I think it should be “anticipated movements” instead of “unanticipated movements”

Line 54-55. “reduced limitations in ankle dorsiflexion range of motion.” What does that mean? Is not KT supposed to constrain ankle motion?

Line 65. Based on the context, it should be “suboptimal position” instead of “optimal position”

Line 81-82. There is insufficient discussion in the introduction to support the authors’ hypotheses. In other words, why did the author hypothesize that KT would reduce peak ankle plantarflexion and inversion angles and moments, and increase peak knee flexion angle?

Line 91. How do you define “engaged”?

Line 110. Based on Figure 1, I think it should be “lateral malleolus.”

Line 135. The speed value (2.7 ± 0.4 m/s) looks like a actual outcome rather than a target value.

Line 138. I would not call it “blinded.” The participants were very well aware of the tasks after signals were given.

Line 156-157. Incorrect punctuation.

Line 198-199. How is the KT method applied in the study able to “reduce gastrocnemius contraction” while also “promote tibialis anterior contraction”?

Line 216-217. Again, it should be “lateral malleolus.”

Line 219. “Poorer function” in what aspect?

Line 229. The authors emphasized many times that lower limb joints were kinematically coupled and the knee joint would supplement restricted ankle motion under intervention conditions. Then here comes a contradictory statement of “Given that the taping did not cover the knee joint, the likelihood of KT improving peak knee flexion angle through mechanical effects is relatively low.”

Line 244. The logic of “limiting a portion of joint ROM while maintaining joint stability[41]” is ambiguous. Suggest to rephrase it.

Line 253-254. How does increased knee flexion “reducing the risk of ACL injuries?”

Line 259. “In addition, the taping retraction from the medial malleolus to the lateral malleolus imparts tension to ankle eversion.” There were no tapes applied in this manner in the study.

Line 267. What is CAI?

All points made in the Limitations sections were off-point. EMG was not the research focus and therefore not supposed to be the limitation. KT is usually not expected to provide sustainable and long-term effects so studying “immediate effects” falls short of being one of the limitations. Non-dominant side also has its own “habitual patterns.” The arguments here are misleading.

Reviewer #2: The article contains a novel and applicable criteria and sounds very good.

Introduction meets the general structure and reflects the importance of the research..

Sampling methods and methodology also are well prepared and well orianted justifications regarding the findings are available.

6. PLOS authors have the option to publish the peer review history of their article (what does this mean?). If published, this will include your full peer review and any attached files.

Reviewer #1: No

Reviewer #2: No

---

## [Author Response · Author response to Decision Letter 0]

6 Apr 2024

Dear Editor

We are very grateful to your comments for the manuscript. These opinions help to improve academic rigor of our article. According with your advice, we tried our best to amend the relevant part and made some changes in the manuscript. Furthermore, we have written a point-by-point response letter for editor and reviewer. You can see the details at “response to reviewers”.

We appreciate for Reviewers’ warm work earnestly, and hope that the correction will meet with approval. Should you have any questions, please contact us without hesitate.

Best wishes to you.

---

## [Decision Letter · Decision Letter 1]

30 Apr 2024

PONE-D-24-02413R1Effects of Ankle Kinesio Taping on Knee and Ankle Joint Biomechanics During Unanticipated Jumps in Collegiate AthletesPLOS ONE

Dear Dr. Chen,

Thank you for submitting your manuscript to PLOS ONE. After careful consideration, we feel that it has merit but does not fully meet PLOS ONE’s publication criteria as it currently stands. Therefore, we invite you to submit a revised version of the manuscript that addresses the points raised during the review process.

We look forward to receiving your revised manuscript.

Kind regards,

Fei Yan

Academic Editor

PLOS ONE

Reviewers' comments:

Reviewer's Responses to Questions

**Comments to the Author**

1. If the authors have adequately addressed your comments raised in a previous round of review and you feel that this manuscript is now acceptable for publication, you may indicate that here to bypass the “Comments to the Author” section, enter your conflict of interest statement in the “Confidential to Editor” section, and submit your "Accept" recommendation.

Reviewer #1: All comments have been addressed

Reviewer #3: (No Response)

2. Is the manuscript technically sound, and do the data support the conclusions?

Reviewer #1: Partly

Reviewer #3: Partly

3. Has the statistical analysis been performed appropriately and rigorously? 

Reviewer #1: Yes

Reviewer #3: Yes

4. Have the authors made all data underlying the findings in their manuscript fully available?

Reviewer #1: Yes

Reviewer #3: Yes

5. Is the manuscript presented in an intelligible fashion and written in standard English?

Reviewer #1: Yes

Reviewer #3: Yes

6. Review Comments to the Author

Reviewer #1: The authors have addressed all my previous comments properly and sufficiently. The manuscript in its current form has advanced to a status that can be considered for publication, on the condition that the following follow-up questions can be

1. Further to my comments to Line 198-199 in the first-round revision:

Based on the citations provided by the authors as well as the relevant research findings in the literature, a taping method is considered to inhibit muscle contraction only when the tapes are applied with tension across the muscle belly, namely perpendicular to the direction of the muscle fibers. According to the authors’ depictions and the figure’s manifestation, the Kinesio tapes were applied in Y shape along the belly of the two gastronomic heads in the current study, which I hardly agree would impose inhibitory effects on muscle contraction.

I suggest revising relevant content to refrain from emphasizing the “inhibitory effects” of the taps applied on gastrocnemius.

2. Further to my comments to Line 253-24 and Line 283-288.

Please attach references to support the corresponding arguments, for example, increasing knee flexion reduces the risk of ACL injury and evidence of gender differences in knee/ankle biomechanics during unanticipated jumping.

Reviewer #3: In this manuscript, the authors investigated the effects of ankle kinesio taping on the kinematic and kinetics of knee and ankle joints during jump tasks in athletes.

1. Lines 46-47: Please update the numbers to more recent references as the currently cited references are 40 and 7 years old, respectively.

2. Lines 74-78: These sentences are confusing and should be removed since the present study did not evaluate the ACL property.

3. The objective of the study should be rewrite and included in a separate paragraph at the end of introduction section. Please conclude with a brief statement of the overall aim of the work and a comment about whether that aim was achieved.

4. Please rewrite and reorganize the last paragraph of the Introduction section.

5. Only 18 subjects were included. The sample size is too small, making the results less reliable when extrapolating to a larger population. While G-Power was used to calculate the sample size, it is assumed that gender does not have an impact on the results. However, if gender does indeed influence the outcomes, the current sample size would be considered inadequate. A larger sample size should be considered to enhance the robustness of the results.

6. Furthermore, it is crucial to investigate potential gender effects and whether the same results can be expected across different gender groups. Including an analysis of gender effects is important to ensure a comprehensive understanding of the outcomes. Additionally, it is assumed that gender does not have an impact on the results in the manuscript. However, if gender does indeed influence the outcomes, the current sample size would be considered inadequate.

7. Figure 1: Please include a figure illustrating the three treatments: NT, PT, and KT, to ensure clarity and facilitate understanding.

8. Figure 2: Please include a figure illustrating the three different jump patterns to provide visual clarity and improve understanding of the experimental design.

9. Please clarify whether all participants performed all three different jump patterns using their right leg. Please include the information in the data collection section.

10. Conclusion section should be improved. The authors should provide a concise summary of the main findings and their significance, as well as potential directions for future research.

7. PLOS authors have the option to publish the peer review history of their article (what does this mean?). If published, this will include your full peer review and any attached files.

Reviewer #1: No

Reviewer #3: No

---

## [Author Response · Author response to Decision Letter 1]

22 May 2024

Summary of a Point-by-Point Response to Reviewers’ Critiques

Manuscript PONE-D-24-02413R1 entitled “Effects of Ankle Kinesio Taping on Knee and Ankle Joint Biomechanics During Unanticipated Jumps in Collegiate Athletes”

Thank you very much for your kindly comments on our manuscript. There is no doubt that these comments are valuable and very helpful for revising and improving our manuscript. In what follows, we would like to answer the questions you mentioned and give detailed account of the changes made to the manuscript. Revised portions are marked yellow in the manuscript. The main corrections in the manuscript and the responses to the reviewers' comments are listed below.

Reviewer #1: 

The authors have addressed all my previous comments properly and sufficiently. The manuscript in its current form has advanced to a status that can be considered for publication, on the condition that the following follow-up questions can be

1. Further to my comments to Line 198-199 in the first-round revision:

Based on the citations provided by the authors as well as the relevant research findings in the literature, a taping method is considered to inhibit muscle contraction only when the tapes are applied with tension across the muscle belly, namely perpendicular to the direction of the muscle fibers. According to the authors’ depictions and the figure’s manifestation, the Kinesio tapes were applied in Y shape along the belly of the two gastronomic heads in the current study, which I hardly agree would impose inhibitory effects on muscle contraction.

I suggest revising relevant content to refrain from emphasizing the “inhibitory effects” of the taps applied on gastrocnemius.

Reply: Thank you for pointing this out. We have removed the reference to gastrocnemius inhibition. The previous content is in lines 200-201.

2. Further to my comments to Line 253-24 and Line 283-288.

Please attach references to support the corresponding arguments, for example, increasing knee flexion reduces the risk of ACL injury and evidence of gender differences in knee/ankle biomechanics during unanticipated jumping.

Reply: Thanks for your valuable counsel. We have added references to knee flexion angle(doi: 10.2519/jospt.2015.5785. doi: 10.1589/jpts.28.2316. Epub 2016 Aug 31.) and gender differences(doi: 10.1080/02640414.2020.1837481. doi: 10.1136/bjsm.2008.055954.). In addition, we performed subgroup analyses of all outcome indicators according to gender differences, and the results support the statement in limitations(Table 3). The results of subgroup analysis of gender differences are in the supporting information, and the file name is Two-way repeated measures ANOVA.

Line 255-257: This can not only decrease the load on lateral ankle ligaments and mitigate the risk of ankle sprains but also may have potential implications for reducing the risk of ACL injuries[42, 43].

Line 285-287: Another limitation is that men and women were not analyzed separately because of the relatively small sample size, despite some evidence of biomechanical differences between sexes[46, 47].

Reviewer #2: 

In this manuscript, the authors investigated the effects of ankle kinesio taping on the kinematic and kinetics of knee and ankle joints during jump tasks in athletes.

1. Lines 46-47: Please update the numbers to more recent references as the currently cited references are 40 and 7 years old, respectively..

Reply: Thanks for your suggestion. We have updated the references and made corresponding revisions in the manuscript(doi: 10.1136/bmjopen-2022-069867. doi: 10.1016/j.jshs.2019.07.009.).

Lines 46-48: Globally, approximately 1 in 10,000 people suffer ankle sprains daily, with 74% of these cases eventually progressing to chronic ankle instability, which poses a substantial healthcare burden[1, 2].

2. Lines 74-78: These sentences are confusing and should be removed since the present study did not evaluate the ACL property.

Reply: Thanks for your reminding. Numerous studies have indicated a correlation between ankle sprains and ACL injuries. As the knee joint is an adjacent joint to the ankle joint, the alterations of ankle biomechanical characteristics can induce corresponding changes in knee biomechanical characteristics. The biomechanical indicators of the knee joint used in this study have also been demonstrated to be associated with the risk of ACL injuries, such as knee valgus angle. Therefore, this study elucidated the risk of ACL injury. As you pointed out, since we did not directly assess ACL attributes, we have downplayed the content related to ACL injuries in the discussion and conclusion.

3. The objective of the study should be rewrite and included in a separate paragraph at the end of introduction section. Please conclude with a brief statement of the overall aim of the work and a comment about whether that aim was achieved.

Reply: Thanks for your valuable counsel. We have rewritten the final paragraph of the introduction according to your suggestion. Additionally, we briefly outline whether this study achieved the expected results in the first paragraph of the discussion.

Line 79-84: The present study employed a randomized signal light system emitting three distinct signals to simulate an unanticipated environment similar to actual movement. We compared the differences in knee and ankle biomechanical characteristics among collegiate athletes under three treatments: no taping (NT), placebo taping (PT), and KT, thus providing a reference for the application of KT in actual sports. The hypothesis of this study was that KT can reduce peak ankle plantarflexion and inversion angles and moments, and increase peak knee flexion angle.

Line 193-195: Partially consistent with the expected results, the present study revealed that ankle KT reduced peak ankle inversion angle, peak vGRF, and peak ankle inversion moment and increased peak knee flexion angle.

4. Please rewrite and reorganize the last paragraph of the Introduction section.

Reply: Thanks for your valuable counsel. We have rewritten the final paragraph of the introduction according to your suggestion.

Line 79-84: The present study employed a randomized signal light system emitting three distinct signals to simulate an unanticipated environment similar to actual movement. We compared the differences in knee and ankle biomechanical characteristics among collegiate athletes under three treatments: no taping (NT), placebo taping (PT), and KT, thus providing a reference for the application of KT in actual sports. The hypothesis of this study was that KT can reduce peak ankle plantarflexion and inversion angles and moments, and increase peak knee flexion angle.

5. Only 18 subjects were included. The sample size is too small, making the results less reliable when extrapolating to a larger population. While G-Power was used to calculate the sample size, it is assumed that gender does not have an impact on the results. However, if gender does indeed influence the outcomes, the current sample size would be considered inadequate. A larger sample size should be considered to enhance the robustness of the results.

Reply: We apologize for this confusion. Due to the limited sample size, we did not analyze male and female participants separately. Therefore, we have accounted for the limitations with the intention of providing some valuable information for future research. We recommend that future research should utilize a larger sample size and conduct analyses based on gender. Moreover, we employed two-way repeated measures ANOVA to recalculate all outcome indicators. The results indicated that there are indeed gender differences in the biomechanical characteristics of participants during unanticipated jumps(Table 3). The results of subgroup analysis of gender differences are in the supporting information, and the file name is Two-way repeated measures ANOVA.

6. Furthermore, it is crucial to investigate potential gender effects and whether the same results can be expected across different gender groups. Including an analysis of gender effects is important to ensure a comprehensive understanding of the outcomes. Additionally, it is assumed that gender does not have an impact on the results in the manuscript. However, if gender does indeed influence the outcomes, the current sample size would be considered inadequate.

Reply: Your suggestion is highly valuable. We also believe it is important to analyze biomechanical characteristics during unanticipated jumps based on gender differences. However, the primary aim of this study was to investigate knee and ankle joint biomechanical characteristics during unanticipated jumps. Therefore, we have accounted for the limitations with the intention of providing some valuable information for future research. We recommend that future research should utilize a larger sample size and conduct analyses based on gender. Moreover, we employed two-way repeated measures ANOVA to recalculate all outcome indicators. The results indicated that there are indeed gender differences in the biomechanical characteristics of participants during unanticipated jumps(Table 3). The results of subgroup analysis of gender differences are in the supporting information, and the file name is Two-way repeated measures ANOVA.

7. Figure 1: Please include a figure illustrating the three treatments: NT, PT, and KT, to ensure clarity and facilitate understanding.

Reply: Thanks for your suggestion. We have replaced the figure 1, and the replaced image can clearly reflect the processing methods under NT, PT, and KT conditions. We have also provided specific instructions in the taping procedure for the three conditions.

8. Figure 2: Please include a figure illustrating the three different jump patterns to provide visual clarity and improve understanding of the experimental design.

Reply: Thank you for your suggestion. Since only left and right jumping maneuvers were analyzed in this study, we provided jump pattern diagrams illustrating the jumping methods for both left and right jumps.

9. Please clarify whether all participants performed all three different jump patterns using their right leg. Please include the information in the data collection section.

Reply: We apologize for the confusion. We have revised the description of the jump patterns accordingly. When participants performed left and right jumps, their right leg landed on the force plate. But for double-leg jumping, both legs landed separately on two force plates.

Line 131-135: When the signal light turned blue, the participants ran and landed on the right force plate with the right leg and quickly jumped to the left force plate. When it turned green, they ran and landed on the left force plate with the right leg and then jumped to the right force plate. When the light was red, the participants ran, landed each leg on two separate plates, and performed a vertical jump (Fig 2)

10. Conclusion section should be improved. The authors should provide a concise summary of the main findings and their significance, as well as potential directions for future research.

Reply: Your suggestion is highly valuable, and we have made the requested revisions accordingly.

Line 289-293: During unanticipated jump maneuvers, KT reduced peak ankle inversion angle, peak vGRF, and peak ankle inversion moment and increased peak knee flexion angle in collegiate athletes. This suggests that KT may have a positive effect on reducing the risk of ankle sprains in healthy individuals. Future research should explore whether KT can also mitigate the risk of ankle sprains in individuals with musculoskeletal disorders under unanticipated conditions.

---

## [Decision Letter · Decision Letter 2]

31 May 2024

Effects of Ankle Kinesio Taping on Knee and Ankle Joint Biomechanics During Unanticipated Jumps in Collegiate Athletes

PONE-D-24-02413R2

Dear Dr. Chen,

We’re pleased to inform you that your manuscript has been judged scientifically suitable for publication and will be formally accepted for publication once it meets all outstanding technical requirements.

Kind regards,

Fei Yan

Academic Editor

PLOS ONE

Additional Editor Comments (optional):

Reviewers' comments:

Reviewer's Responses to Questions

**Comments to the Author**

1. If the authors have adequately addressed your comments raised in a previous round of review and you feel that this manuscript is now acceptable for publication, you may indicate that here to bypass the “Comments to the Author” section, enter your conflict of interest statement in the “Confidential to Editor” section, and submit your "Accept" recommendation.

Reviewer #1: All comments have been addressed

Reviewer #3: All comments have been addressed

2. Is the manuscript technically sound, and do the data support the conclusions?

Reviewer #1: Yes

Reviewer #3: Partly

3. Has the statistical analysis been performed appropriately and rigorously? 

Reviewer #1: Yes

Reviewer #3: Yes

4. Have the authors made all data underlying the findings in their manuscript fully available?

Reviewer #1: Yes

Reviewer #3: Yes

5. Is the manuscript presented in an intelligible fashion and written in standard English?

Reviewer #1: Yes

Reviewer #3: Yes

6. Review Comments to the Author

Reviewer #1: All my previous comments have been addressed properly and sufficiently. The current form of the manuscript meets publication standards.

Reviewer #3: The authors have addressed the concerns raised in the previous review after a thorough review of the manuscript.

7. PLOS authors have the option to publish the peer review history of their article (what does this mean?). If published, this will include your full peer review and any attached files.

Reviewer #1: No

Reviewer #3: No
